# Epidemics of HIV Infection among Heavy Drug Users of Depressants Only, Stimulants Only, and Both Depressants and Stimulants in Mainland China: A Series, Cross-Sectional Studies

**DOI:** 10.3390/ijerph17155483

**Published:** 2020-07-29

**Authors:** Bo Zhang, Xiang-Yu Yan, Yong-Jie Li, Zhi-Min Liu, Zu-Hong Lu, Zhong-Wei Jia

**Affiliations:** 1Department of Epidemiology and Biostatistics, School of Public Health, Peking University, Beijing 100191, China; bibibabo@pku.edu.cn (B.Z.); yanxiangyu@bjmu.edu.cn (X.-Y.Y.); li.yongjie@outlook.com (Y.-J.L.); 2National Institute on Drug Dependence, Peking University, Beijing100191, China; zhiminliu@bjmu.edu.cn; 3Biomedical Engineering, Southeast University, Nanjing 211189, China; zhlu@seu.edu.cn; 4Center for Drug Abuse Control and Prevention, National Institute of Health Data Science, Peking University, Beijing 100191, China; 5Center for Intelligent Public Health, Institute for Artificial Intelligence, Peking University, Beijing 100191, China

**Keywords:** HIV infection, epidemics, stimulants, depressants, injection drug use, China

## Abstract

*Background:* Heavy drug users was a global consensus high-risk population of HIV infection. However, the specific impact of drug on HIV infection has not yet been established. Depressants and stimulants were most widely used drugs in mainland China, and mix use of the two drugs was also serious. We assessed the HIV infection rate and trends in heavy drug users by analyzing data from the National Dynamic Management and Control Database for Drug Users (NDMCDDU). *Methods:* All heavy drug users with HIV test results in NDMCDDU from 2008 to 2016 were grouped into depressants only group (DOG), stimulants only group (SOG), and both depressants and stimulants group (DSG). We used joinpoint regression to examine trends of HIV infection rates. Multivariable logistic regression was used to examine factors related to HIV infection. *Results:* A total of 466,033 heavy drug users with 9522 cases of HIV infection were included in this analysis. HIV infection rate was estimated at 2.97% (95% CI 2.91–3.04%) of 265,774 users in DOG, 0.45% (95% CI 0.42–0.49%) of 140,895 users in SOG, and 1.65% (95% CI 1.55–1.76%) of 59,364 users in DSG. In DOG, a U-shaped curve of HIV infection rate decreased from 3.85% in 2008 to 2.19% in 2010 (annual percent change (APC) −12.9, 95% CI −19.3–−6.0, *p* < 0.05), then increased to 4.64% in 2016 (APC 8.3, 95% CI 6.1–10.4, *p* < 0.05) was observed. However, SOG and DSG showed consistent increases from 0.15% in 2008 to 0.54% in 2016 (APC 8.2, 95% CI 4.8–11.8, *p* < 0.05) and from 0.78% in 2008 to 2.72% in 2016 (APC 13.5, 95% CI 10.7–16.4, *p* < 0.05), respectively. HIV infection rate of DOG in the southwest region presented a U-shaped trend. All groups showed significant increases in HIV infection in east and central regions. *Conclusions:* The U-shaped curve for HIV infection rate among DOG users and consistent increases among SOG and DSG users implies drug abuse is still a critical focus of HIV infection in China. It is urgently needed to reassess the effectiveness of current strategies on HIV prevention and control among drug users.

## 1. Introduction

Drug users was a global consensus high-risk population of HIV infection [1]. Especially heavy drug users, who endured more severe health burden such as HIV and other blood-borne infectious diseases [2]. The studies of the “pure” heavy drug users (who uses one specific drug type exclusively) would provide information on the relationship between drug and HIV infection [3]. However, the specific impact of “pure” drug on HIV infection has not yet been established in China, because heavy drug use is usually accompanied by the use of multiple drugs [4].

In mainland China, depressants (opiates and synthetic opioids) and stimulants (amphetamines, methamphetamine, etc.) were major widely used drugs, which accounted for 93% of all drug users in 2019 [5]. Depressants, were traditionally, the most abused drugs. The majority of depressants users were young (30 years or younger), non-married (60%), and males (60–70%) with low education levels and without stable jobs [6]. In southwest regions, with a high prevalence of depressants use (Yunnan, Sichuan, Gansu, and Guangdong provinces), the rate of intravenous administration in depressants users was about 1/3 [6]. Intravenous administration suggested heavy drug use, and sharing contaminated syringes could make heavy drug users endure the greatest health risks, more than one in eight among them live with HIV [1]. Prior to 2005, persons who inject drugs (PWID), mainly depressants users, were predominantly responsible for new HIV/AIDS infections [7]. However, thanks to the expansion of harm reduction programs, such as methadone maintenance treatment (MMT) and Needle and Syringe Programs (NSP), the national HIV prevalence among PWID decreased slightly from 10.6% in 2002 to 9.1% in 2010 [8,9,10].

The number of stimulants users has exceeded depressants users since 2014, and they have become the most widely used drugs in mainland China [11]. The abuse of stimulants has spread throughout the country, with concentrations in southeastern regions and business-centered cities [6,12]. More and more company staffers, entertainers, and students are becoming stimulants users [6,12]. Different from heavy depressants users, Shu Su reported more heavy stimulants users were female, younger, single and higher monthly income [13]. Stimulants are powerful central nervous system drugs associated with high-risk sexual practices and chem-sex addiction [14,15]. Accordingly, unprotected sexual behaviors increase the risk of HIV transmission among stimulants users [16,17]. This suggested the influence of drugs on HIV transmission is no longer limited in injection drug user (IDU), drugs have potentially linked high risk populations of HIV infection [16,18]. Yanming Sun’s study found that stimulants abuse appeared to be correlated with commercial sexual behaviors and high HIV prevalence among female drug users [16]. Xu JJ reported that stimulants abuse was popular among Chinese men who have sex with men (MSM) and was associated with significant increased HIV infection risk [18].

In 2002, China issued “Standards for HIV Surveillance” to standardize the practice of HIV surveillance [19]. The approach utilized surveillance sites for drug users over the country and sampled 400 people twice each year from every surveillance site to estimate the prevalence, incidence and risk behaviors of HIV infection [19]. Similarly, global researches on nationwide HIV epidemic of drug users are also based on surveillance data. In 2003, Centers for Disease Control and Prevention (CDC) in U.S. created National HIV Behavioral Surveillance (NHBS) to conduct behavioral surveillance among persons at high risk for HIV infection in 23 cities [20]. However, researches on national HIV epidemic of drug users based surveillance data have some limitations. First, self-reported drug abuse behavior has potential report bias, especially underestimated the number of depressants users (heroin, etc.) who are more hard-to-reach as penalty is more severe [21]. Second, data such as drug types investigated through questionnaires of sentinel surveillance sites are not precise as expected because the answers will be greatly affected by the subjective consciousness of the respondents. Last, there are more than 660,000 villages (smallest administrative unit) in China, the number is far more than the number of surveillance sites for drug users, thus, the representativeness of sample data collected by surveillance sites is limited [22].

China has set up National Dynamic Management and Control Database for Drug Users (NDMCDDU) to register nationwide drug users [23]. In NDMCDDU, drug types were verified by urine tests, which were more accurate than data investigated by sentinel surveillance sites [19,24]. In addition, data in the NDMCDDU is collected by Government staff, which covered all the smallest administrative units and recorded all observed drug users in mainland China [22]. We presume that the database is representative and such a work of evaluate a real database is of great value and helpful for prevention and control on drug and infectious diseases in real public issues. It may give some policy implications for the Chinese Government and other countries in the world. Therefore, in this study, we aim to assess nationwide HIV infection trends and to examine socio-demographic and geographic characteristics of HIV infection among heavy drug users in mainland China from 2008 to 2016 by analyzing data in the NDMCDDU.

## 2. Materials and Methods

### 2.1. Definitions

The following terms are used in this study.

Depressants: drugs recorded in NDMCDDU includes heroin, opium, poppy capsule, morphine, pethidine hydrochloride, dihydroetorphine, buprenorphine, and tramadol.

Stimulants: drugs recorded in NDMCDDU includes ecstasy, methamphetamine and amphetamine.

Drug users: those who used prohibited depressants and stimulants for non-medical purposes and observed by the Government.

Heavy drug users: according to Chinese “Measures for Drug Addiction”, drug users who injected drugs or abused drugs multi-times for non-medical use were judged by professionals as heavy drug users. Heavy drug users shall be tested for HIV and recorded in NDMCDDU before entering detoxification centers for treatment [25].

Depressants only group (DOG): included drug users who had been recorded using depressants only.

Stimulants only group (SOG): included drug users who had been recorded using stimulants only.

Depressants and stimulants group (DSG): included drug users who had been recorded as using depressants and stimulants but not any other type drugs.

PWID: self-reported injection drug use and recorded in NDMCDDU.

HIV infection rate: calculated as the number of newly tested and observed HIV-positive drug users (numerator) divided by the number of newly tested drug users in the same year (denominator) recorded in NDMCDDU.

### 2.2. Study Design and Procedures

NDMCDDU is a national registry database set up by the Chinese Government to register drug users, covering all administrative units in the 31 provinces of mainland China over the whole country. All drug users in NDMCDDU were those who have been found using illicit drugs for non-medical purposes and registered by the Government. As of 2016, NDMCDDU recorded about 4 million drug users historically observed in mainland China, which provided strong support for studies on nationwide drug users. Since 2008, the Government started to register HIV test results of heavy drug users in NDMCDDU [24]. Therefore, we assessed all data we have registered in NDMCDDU from 1 January 2008 to 30 June 2016. Socio-demographic characteristics (sex, age, ethnicity, education, and marital status), registered date, location, drug types, methadone treatment history, and HIV status of these drug users were recorded in NDMCDDU, and were extracted directly from the database. The drug types recorded in the database were verified by urine tests. Referring to the “Diagnostic criteria for HIV/AIDS”, the diagnosis of HIV infection was conducted by professional medical institutions through serological screening test and confirmation test [26]. For health concern, HIV-positive drug users who meet the Chinese national treatment criteria (WHO stage 3 or 4 disease or CD4 count of 350 cells per μL or less) are referred for treatment with standard three-drug therapy. In addition to these structured variables, injection drug use behavior was extracted from the text record document by the automatic keyword-based matching technique (details in Appendix A). To protect the privacy of drug users, the data used for analysis given by the Government were anonymized. Names and other individually identifiable information of these drug users were not included in the data, only the ID numbers were used as the unique identification code.

Drug users were classified into DOG, SOG, and DSG according to the drug types they used. Data for individuals with a history of using drugs other than depressants or stimulants were excluded. Primary outcomes were HIV infection trends of the three groups from 2008 to 2016. Secondary outcomes were sociodemographic and geographic characteristics associated with HIV infection.

### 2.3. Measures

Socio-demographic characteristics. Socio-demographic characteristics were registered in NDMCDDU by Government staff when drug users entered detoxification centers, including sex (male and female), ethnicity (Han and minority), education (primary school or no schooling, junior high school, and high school or above), marital status (divorced or widowed, married, and unmarried), year of HIV test and birth in database. The age was measured as interval years between year of birth and first HIV test in NDMCDDU, and we coded age as 4 levels (≤17, 18~24, 25~44, and ≥45 years).

Geographic characteristics. We categorized regions where drug users done HIV test as 7 areas based on “The Physical Geography of China”, including Northeast (Heilongjiang, Jilin, and Liaoning), North (Inner Mongolia, Shanxi, Hebei, Beijing, and Tianjin), East (Shandong, Jiangsu, Zhejiang, Shanghai, Fujian, Jiangxi, and Anhui), Central (Henan, Hubei, and Hunan), South (Guangxi, Guangdong, and Hainan), Southwest (Yunnan, Xizang, Sichuan, Chongqing, and Guizhou), and Northwest (Xinjiang, Gansu, Ningxia, Qinghai, and Shaanxi) [27].

Drug use related variables. We compared methadone treatment and HIV test year in order to determine if the drug user received methadone treatment or not before HIV test (yes or no). Injection drug use or not (yes or no) were extracted directly from the database.

### 2.4. Statistical Analysis

Socio-demographic characteristics and drug use related variables of the three groups were compared by chi-squared tests for categorical variables and one-way ANOVA for continuous variable (age). Logistic regression models were used to compare unadjusted and adjusted odds ratios (ORs and AORs, respectively) and 95% CIs of HIV infection rate among the three groups and by characteristics. Joinpoint regression was used to examine the changing trend of HIV infection rate among the three groups across the country and regions during the study period. Annual percent change (APC) for each line segment and the corresponding 95% confidence interval (95% CI) were estimated. The APC is tested to determine whether a difference exists from the null hypothesis of no change (0%). Each joinpoint informs a statistically significant change in trends (increase or decrease) and each of trends is described by an APC [28].

A two-sided *p* value of 0.05 or less was regarded as significant. Data were checked in PostgreSQL 9.3 (The PostgreSQL Global Development Group, open source database) and SAS version 9.4 (SAS Campus Drive, Cary, NC, USA). Statistical analyses were carried out using SPSS version 22.0 (IBM Corp., Armonk, NY, USA), SAS version 9.4 (SAS Campus Drive, Cary, NC, USA) and Joinpoint Regression Program 4.6.0 (U.S. National Cancer Institute, Bethesda, MD, USA). Geographic visualization was done with ArcGIS 10.0 (Esri Corp, Redlands, CA, USA).

### 2.5. Ethical Issues

The data we have was anonymized by the Government to protect the privacy of drug users, so this study focused on population-level analyses only and did not access any individually identifiable data. Thus, after the assessment by Institutional Review Board, ethical approval was not sought. At the time of registration, drug users were informed and agreed that their information would be registered in database for management and research.

## 3. Results

After excluding 1371 users who had used drugs other than depressants and stimulants, a total of 466,033 heavy drug users were tested for HIV between Jan, 2008 and Jun, 2016. Among which 265,774 (57.0%), 140,895 (30.2%), and 59,364 (12.7%) users were classified as DOG, SOG, and DSG respectively (Figure 1, Appendix B). Drug users included in this study aged 34.3 (SD = 8.6) years. Most users were male (87.9%), Han (83.1%), had junior high school and below education (85.6%). Over half of these drug users were in East (25.3%) and South (25.8%). Compared with DOG and DSG, SOG users tended to be more female (16.7%, *p* < 0.05), younger (30.8 ± 8.3 years), junior school and above education (82.9%) and non-PWID (99.5%) (Table 1). The HIV infection rates of DOG, SOG and DSG users were 2.97% (95% CI 2.91–3.04%), 0.45% (95% CI 0.42–0.49%) and 1.65% (95% CI 1.55–1.76%) respectively (Table 2).

### 3.1. Trends and Socio-Demographic Characteristics of HIV Infection

The HIV infection rate of DOG users presented a U-shaped trend which decreased from 3.85% in 2008 to 2.19% in 2010 (APC −12.9, 95% CI −19.3–−6.0, *p* < 0.05), then increased to 4.64% in 2016 (APC 8.3, 95% CI 6.1–10.4, *p* < 0.05). While the HIV infection rates of SOG and DSG users both maintained increased trends. In SOG, the infection rate increased from 0.15% in 2008 to 0.54% in 2016 (APC 8.2, 95% CI 4.8–11.8, *p* < 0.05). In DSG, the infection rate increased from 0.78% in 2008 to 2.72% in 2016 (APC 13.5, 95% CI 10.7–16.4, *p* < 0.05) (Table 2, Figure 2).

DOG users were more likely to be HIV infected than SOG (AOR 2.07, 95% CI 1.88–2.28, *p* < 0.05) and DSG (AOR 1.28, 95% CI 1.19–1.37, *p* < 0.05) users. Almost, in all subgroups stratified by characteristics, DOG users had higher risk of HIV infection than SOG and DSG users. While there was no statistical difference of HIV infection rate among people in the Northwest or people who have high school or above education in three groups (Table 2).

Among DOG users, females were associated with higher odds ratio of HIV infection (AOR 1.17, 95% CI 1.08–1.26, *p* < 0.05), the same as among SOG (AOR 1.34, 95% CI 1.09–1.66, *p* < 0.05) and DSG (AOR 1.55, 95% CI 1.29–1.86, *p* < 0.05) users. In DOG, a higher proportion of people aged between 25 and 44 were HIV infected compared with adolescents aged under 17 (AOR 3.78, 95% CI 2.07–6.89, *p* < 0.05). While more HIV infections were among aged 45 years older in SOG (AOR 7.20, 95% CI 1.73–29.89, *p* < 0.05) and DSG (AOR 3.04, 95% CI 2.10–4.39, *p* < 0.05). Compared with Han, minorities in DOG (AOR 2.87, 95% CI 2.72–3.02, *p* < 0.05), SOG (AOR 1.68, 95% CI 1.29–2.20, *p* < 0.05) and DSG (AOR 1.87, 95% CI 1.56–2.25, *p* < 0.05) were associated with higher odds ratios for HIV infection. Lower education suggested more HIV infections, individuals who had primary school or below education were associated with higher odds ratio of HIV infection than individuals had high school or above education in DOG (AOR 1.89, 95% CI 1.70–2.09, *p* < 0.05), SOG (AOR 1.35, 95% CI 1.03–1.78, *p* < 0.05) and DSG (AOR 1.97, 95% CI 1.50–2.58, *p* < 0.05). Unmarried people had lower HIV infection rate among DOG (AOR 0.83, 95% CI 0.79–0.87, *p* < 0.05) and DSG (AOR 0.82, 95% CI 0.72–0.95, *p* < 0.05) users. PWID in DOG (AOR 4.96, 95% CI 4.58–5.37, *p* < 0.05), SOG (AOR 7.42, 95% CI 4.79–11.48, *p* < 0.05) and DSG (AOR 2.68, 95%CI 2.28–3.17, *p* < 0.05) endured a higher risk of HIV infection compared with non-PWID. DOG users who received MMT before the HIV test had a higher HIV infection rate compared to individuals who did not receive treatment (AOR 1.80, 95%CI 1.69–1.91, *p* < 0.05) (Table 3).

### 3.2. Geographic Trends in HIV Infection Rate

The HIV infection rate of DOG users in the Southwest was 5.93% (95% CI 5.74–6.13%), higher than other regions except for the Northeast (Table 2 and Table 3). The rate in Southwest presented a U-shaped trend, decreased from 12.00% in 2008 to 4.01% in 2011 (APC −17.2, 95% CI −25.2–−8.3, *p* < 0.05), then increased to 7.38% in 2016 (APC 5.4, 95% CI 0.7–10.2, *p* < 0.05) (Table 3, Figure 3). The HIV infection rate in four regions presented increased trends, which were South (APC 10.3, 95% CI 6.0–14.9, *p* < 0.05), North (APC 6.5, 95% CI 2.6–10.7, *p* < 0.05), East (APC 12.9, 95% CI 8.2–17.8, *p* < 0.05) and Central (APC 46.3, 95% CI 27.7–67.6, *p* < 0.05), respectively. While in the Northwest, the HIV infection rate maintained a decreased trend from 6.36% in 2008 to 2.39% in 2016 (APC −7.2, 95% CI −10.9–−3.3, *p* < 0.05) (Table 2, Figure 3).

The HIV infection rate of SOG users in the Northwest (1.22% (95% CI 0.83–1.74%)) was highest. However, East and Central regions maintained increased trends. Increases from 0.08% in 2008 to 0.30% in 2016 (APC 8.1, 95% CI 3.7–12.8) were observed in the East, and from 0.15% in 2009 to 0.40% in 2016 (APC 16.1, 95% CI 6.8–26.3) were observed in Central, respectively (Table 2, Figure 3).

Among DSG users, the HIV infection rate in the Southwest (4.24% (95%CI 3.78–4.73%)) was highest. In the South, the HIV infection rate increased from 1.36% in 2009 to 3.68% in 2016 (APC 11.2, 95% CI 7.5–15.0). Furthermore, increased trends were also observed in Central (APC 19.2, 95% CI 8.6–30.9) and East regions (APC 14.4, 95% CI 8.8–20.4) (Table 2, Figure 3).

## 4. Discussion

In this study, we extracted HIV data from NDMCDDU between 2008 and 2016, which covered all registered heavy drug users in 31 provinces in mainland China. Our data showed that depressants and stimulants were most commonly used drugs in mainland China, it is meaningful to focus on the health issues of population used these two types of drugs (Table 1, Appendix B
Table A1). We noted strikingly accelerated HIV infection rates among DOG, SOG and DSG users since 2010, although the HIV infection rate of DOG users decreased from 2008 to 2010. DOG users were associated with highest HIV infection rate than DSG and SOG users, this mainly attributed to the intravenous administration of drugs. A larger proportion (32.8%) of DOG users were PWID followed by 25.7% of DSG users, compared with 0.5% of SOG users, so unsafe injecting practices relating to the sharing of contaminated needles and syringes was the main cause of HIV infection [7,8,9]. In our study, the similarity between DOG and DSG users was reflected in the distribution of IDU of drugs and sociodemographic characteristics, and it is reported that depressants users have shifted to stimulants [6]. PWID are a key population at increasing risk of HIV infection around the world. In our study, the HIV infection rate of PWID among DOG users was 5.3%, lower than 17.8% of the global HIV infection rate among PWID, 28.5% in South-West Asia and 24.0% in Eastern and South-Eastern Europe [29,30]. In the Russian Federation, the prevalence of HIV among depressants users (especially in registered PWID) rose steadily over the period 2009–2014, from 13.2% to 19.9%, which was consistent with the increase trend of DOG and DSG in China after 2010 [30]. While in developed countries such as the USA, the HIV infection rate among this population had decreased from 12% in 2006 to 6% in 2018 [31]. In addition, increasing HIV epidemics were reported among stimulants users in different populations in the USA, Russia and other countries, and a similar trend is also observed in our study [32,33].

In China, prior to 2007, injection depressants use was predominantly responsible for new HIV/AIDS infection cases [6,12]. Therefore, China has scaled up harm reduction programs such as MMT and NSP to address this issue, which might lead to the decrease of HIV infection rate before 2010 in DOG [6,12]. However, the limitations of these programs’ implementation in recent years could be a potential reason of the rebound in HIV infection rate of DOG. Though MMT can reduce the risk of HIV transmission by reducing needle sharing and improving self-protection awareness, currently, there are still some problems affecting the effect of MMT treatment in China, such as low treatment coverage, serious discontinuation of treatment and short duration time of treatment [34,35]. Furthermore, in our study, drug users who received MMT were associated with a higher HIV infection rate than those who had not received MMT. This may be attributed to the fact that people receiving MMT were addicted to drugs, more heavily. Heavier addiction may increase the risk of HIV infection due to long-term use of drugs and high frequency of IDU [2,3,4]. These issues posed challenges to MMT’s effect of HIV control, recently, and the effect of the NSP is also elusive. High coverage of needle-syringe programs (246 needles and syringes per person who injects drugs per year) have been reported in China, but the coverage might be overestimated because those who had not yet attended the program were not included in the statistics [36]. Further, NSP has not been extensively evaluated to explore factors influencing acceptability and feasibility [37]. Lei Zhang et al. suggested that continued law enforcement and mandatory detoxification remain as major barriers to the necessary program scale-up and may even counteract the benefits of NSPs [38]. Our study observed the increasing trend of HIV infection rate among depressants only users after 2010, which implied that the promotion and effectiveness of nationwide harm reduction programs needs to be systematically evaluated.

In our study, a higher proportion of HIV infection among heavy drug users was associated with being female, older, racial minorities and undereducated. Females move faster than males towards drug addiction and have a greater vulnerability than males to HIV and other blood-borne infections [6]. Females also have less access to healthcare services to address drug-related health needs, and they have to face the combined stigma of their gender and their status as drug users, including discrimination, in accessing healthcare services [6]. In addition, high HIV infection rates among female drug users may be partly due to the phenomenon of “sex exchange for drugs”. Multiple sexual partners, low condom use, and prevalent sexually transmitted diseases (STDs) have been observed among female drug users in China [16]. According to the latest report on China’s drug situation, the proportion of young people among newly discovered drug users has decreased, but the proportion of drug users over 60 years old has increased by 3.5%, the similar trend has also been observed in Western countries [5]. Longer terms of using drugs among aging drug users and their tendency to have condom-less sex because of the less risk of pregnancy might increase their risk of HIV infection [39]. Racial minorities mainly settled in southwest China, which lags behind on life expectancy and per-capita GDP, faces disproportionately greater HIV vulnerabilities, due in part to high HIV prevalence caused by IDU, and reportedly greater practices of sexual concurrency and inconsistent condom use [40]. It has also been reported that undereducated (OR = 2.32, 95% CI 1.02–5.25) drug users contributed substantially to new HIV infections [41].

From a spatial perspective, most HIV cases among DOG and DSG users were still concentrated in South and Southwest regions initially affected by the epidemic, where situated between the two largest heroin producing regions in the world, the “Golden Crescent” and the “Golden Triangle”. There is a big gap between the western regions (Northwest and Southwest regions) and the eastern (North, East, Northeast and South regions) and central (Central region) regions in terms of the economic situation, income level, technological development level and industrial structure. By the end of 2019, the ratio of per-capita GDP of eastern region, dividing central region and western region was 1.65 and 1.71, respectively. In addition, western regions feature racial minorities inhabiting the districts. The differences in human geography and socio-economic level make people in western regions vulnerable to HIV infection and lead to the imbalance of drug and HIV epidemic in China [42]. Since 2005, western regions have been the focus of the “People’s War on Drugs” in China, and the war has achieved remarkable success, the proportion of heroin users decreased by 52.3% among new drug users between 2003 and 2010, and the HIV prevalence of PWID decreased from 10.6% in 2002 to 9.1% in 2010 [9,43]. But in our study, since 2010, HIV infection rate among DOG users in other regions almost all maintained increased trends, the rise of HIV infection rate requires constant vigilance. The significantly sharpest increased trends of HIV infection rate among stimulants users were observed in East and Central regions, and our study showed that the increased trends were consistent with substantial increases of stimulants users in East and Central regions, where contain population densities of >450 people per square kilometer and account for about 46% of China’s population [44]. This finding implies that HIV infection is easily spreading to the general population, which highlighted the new challenge of HIV epidemic.

Our study has several limitations. First, people tested for HIV are heavy drug users, therefore, there is a lack of information about drug users with mild addiction. Second, self-reported drug use patterns might contain report bias and cause the underestimation of the number of PWID. Last, this study was a secondary data analysis, data on sexual behavior were not available, and we were unable to adjust for this risk factor. Nevertheless, to our knowledge, this is the first estimate of HIV infection rate among the largest population of heavy drug users, nationwide. The findings of our study highlight implications of public health policy for HIV prevention and control of heavy drug users. First of all, given the expanding of HIV epidemic among heavy depressants users, the Government needs to scale up the coverage of NSP and MMT to prevent the development of drug addiction and reduce the risk of HIV transmission, especially in central and east regions. Further, adequate assessment system and methods are urgently needed to be constructed to evaluate the effectiveness of harm reduction programs. Second, drug use and HIV prevention, treatment and care should be tailored to the specific needs of vulnerable populations such as female, older and undereducated people. Another issue to note is that the global COVID-19 pandemic has plunged the world into an unprecedented crisis. Restrictions on movement and gatherings put in place to halt the spread of COVID-19, may lead to an overall decrease in consumption of recreational drugs. However, drug supply shortages lead to an increase in prices, and some users began seeking out cheaper synthetic substances, and thus the patterns of use shifted towards injecting drugs. Therefore, governments should not reduce drug-related budgets, and also consider especially, the delivery of drug treatment and care services. What is more, during the COVID-19 pandemic, attention should be paid to the provision of routine medical services and drug supply to HIV infected persons, to ensure continuity of HIV prevention and treatment services.

## 5. Conclusions

The U-shaped curve for HIV infection rate among DOG users and consistent increases among SOG and DSG users implies heavy drug use is a critical focus of HIV infection in China. Our results suggest that we should still focus on the continuous rise of HIV infection rate of heavy drug users. Drug use and HIV prevention, treatment and care, should be tailored to the specific needs of vulnerable populations such as females, older people, racial minorities, undereducated and non-married persons, who are those living with HIV and heavily using drugs. Although most HIV cases were still concentrated in western provinces, initially affected by the drug abuse and HIV epidemic, HIV infection rate among heavy drug users in Central and East regions all maintained increased trends. Therefore, we urgently need to reassess the effectiveness of current strategies on HIV prevention and control among heavy drug users and eliminate discrimination and unfair distribution of health resources regarding gender, social status and geographic inequality.

## Figures and Tables

**Figure 1 ijerph-17-05483-f001:**
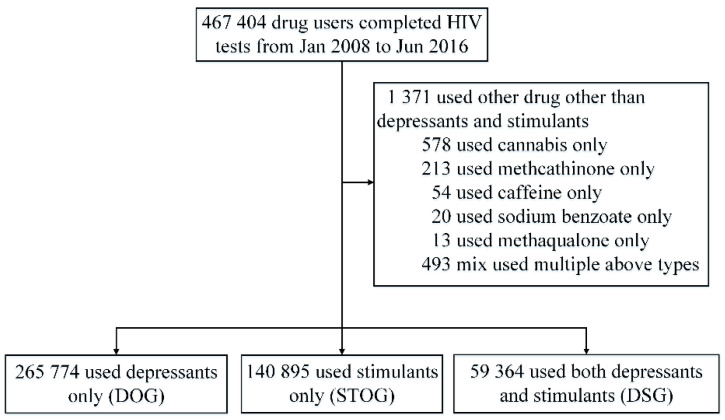
Group classification.

**Figure 2 ijerph-17-05483-f002:**
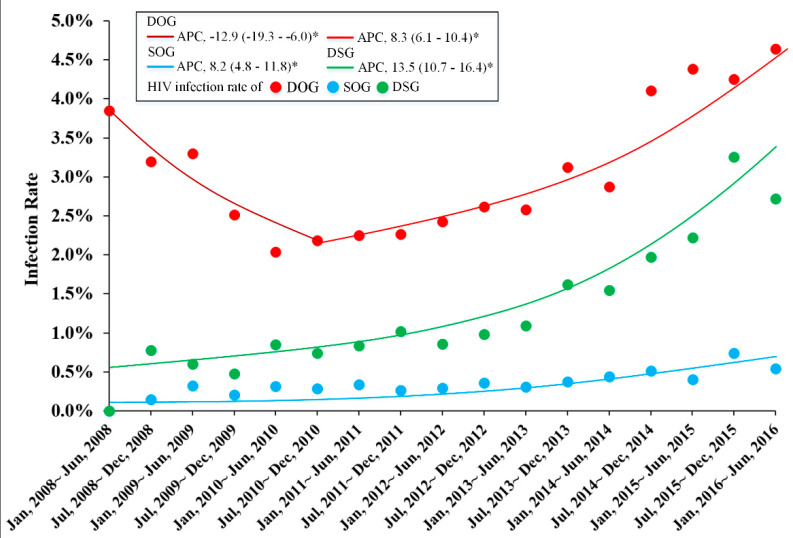
Trends of HIV infection rate among depressants only, stimulants only and both depressants and stimulants users. DOG: depressants users. SOG: stimulants users. DSG: both depressants and stimulants users. * suggested *p* < 0.05.

**Figure 3 ijerph-17-05483-f003:**
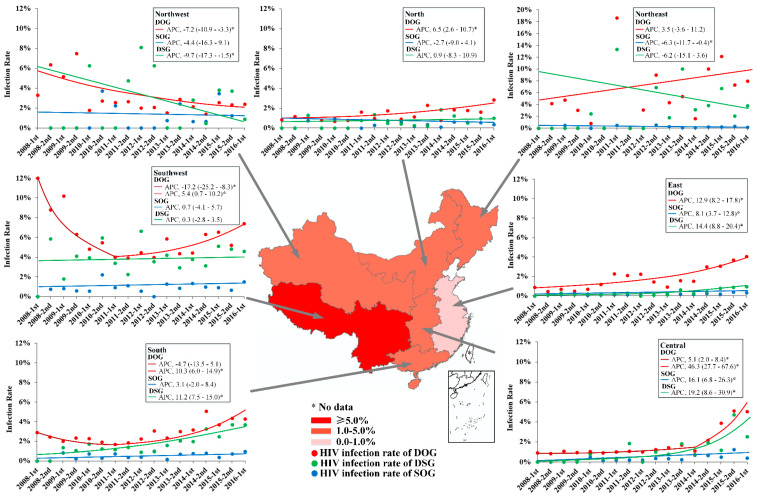
Trends of HIV infection rate among depressants only, stimulants only and both depressants and stimulants users stratified by regions. DOG: depressants only users. SOG: stimulants only users. DSG: both depressants and stimulants users. 1st: the first half year from January to June. 2nd: the second half year from July to December. * suggested *p* < 0.05.

**Table 1 ijerph-17-05483-t001:** Characteristics of depressants only users, stimulants only users and both depressants and stimulants users in China, 2008–2016.

Characteristics	Total	DOG	SOG	DSG	*p*-Value
	N (%)	N (%)	N (%)	N (%)	
**Total**	466,033	265,774	140,895	59,364	
**HIV** ^a,b,c^					<0.001
Positive	9522 (2.0)	7906 (3.0)	635 (0.5)	981 (1.7)	
Negative	456,511 (98.0)	257,868 (97.0)	140,260 (99.5)	58,383 (98.3)	
**Sex** ^a,b,c^					<0.001
Male	409792 (87.9)	240,730 (90.6)	117,297 (83.3)	51,765 (87.2)	
Female	56241 (12.1)	25,044 (9.4)	23,598 (16.7)	7599 (12.8)	
**Age**^a,b,c,d^ (IQR)	34 (28–40)	35 (29–41)	29 (25–36)	36 (30–42)	<0.001
**Ethnicity** ^a,b,c^					<0.001
Han	387,488 (83.1)	215,647 (81.1)	119,358 (84.7)	52,483 (88.4)	
Minority	46,698 (10.1)	35,014 (13.2)	7485 (5.3)	4199 (7.1)	
Missing	31,847 (6.8)	15,113 (5.7)	14,052 (10.0)	2682 (4.5)	
**Education** ^a,b,c^					<0.001
High school or above	54,002 (11.6)	24,874 (9.4)	21,977 (15.6)	7151 (12.0)	
Junior high school	287,312 (61.6)	154,243 (58.0)	94,789 (67.3)	38,280 (64.5)	
Primary school or below	111,886 (24.0)	77,349 (29.1)	21,666 (15.4)	12,871 (21.7)	
Missing	12,833 (2.8)	9308 (3.5)	2463 (1.7)	1062 (1.8)	
**Marital status** ^a,b,c^					<0.001
Married	201,101 (43.2)	100,079 (37.7)	76,786 (54.5)	24,236 (40.8)	
Unmarried	217,183 (46.6)	135,569 (51.0)	52,955 (37.6)	28,659 (48.3)	
Divorced or Widowed	38,190 (8.1)	22,748 (8.6)	9592 (6.8)	5850 (9.9)	
Missing	9559 (2.1)	7378 (2.8)	1562 (1.1)	619 (1.0)	
**Injection drug use** ^a,b,c^					<0.001
No	146,537 (31.4)	59,374 (22.3)	65,330 (46.4)	21,833 (36.8)	
Yes	103,117 (22.1)	87,142 (32.8)	726 (0.5)	15,249(25.7)	
Not sure	216,379 (46.4)	119,258 (44.9)	74,839 (53.1)	22,282 (37.5)	
**Methadone treatment** ^a,b,c^					<0.001
No	419,313 (90.0)	229,374 (86.3)	140,895 (100.0)	50,169 (84.5)	
Yes	46,720(10.0)	36,400 (13.7)	0(0.0)	9195 (15.5)	
**Date of HIV test** ^a,b,c^					<0.001
Jan, 2008~Jun, 2008	2266 (0.5)	2078 (0.8)	116 (0.1)	72 (0.1)	
Jul, 2008~Dec, 2008	24,754 (5.3)	21,040 (7.9)	2045 (1.5)	1669 (2.8)	
Jan, 2009~Jun, 2009	27,565 (5.9)	24,030 (9.0)	1877 (1.3)	1658 (2.8)	
Jul, 2009~Dec, 2009	31,843 (6.8)	24,894 (9.4)	4416 (3.1)	2533 (4.3)	
Jan, 2010~Jun, 2010	27,677 (6.0)	20,936 (7.9)	4150 (2.9)	2591 (4.4)	
Jul, 2010~Dec, 2010	24,673 (5.3)	18,062 (6.8)	4176 (3.0)	2435 (4.1)	
Jan, 2011~Jun, 2011	22,418 (4.8)	16,181 (6.1)	3846 (2.7)	2391 (4.0)	
Jul, 2011~Dec, 2011	19,075 (4.1)	13,425 (5.1)	3388 (2.4)	2262 (3.8)	
Jan, 2012~Jun, 2012	22,412 (4.8)	14,519 (5.5)	5091 (3.6)	2802 (4.7)	
Jul, 2012~Dec, 2012	21,898 (4.7)	13,360 (5.0)	5595 (4.0)	2943 (5.0)	
Jan, 2013~Jun, 2013	26,754 (5.7)	14,420 (5.4)	8207 (5.8)	4127 (7.0)	
Jul, 2013~Dec, 2013	31,350 (6.8)	16,439 (6.2)	10,473 (7.4)	4438 (7.5)	
Jan, 2014~Jun, 2014	31,597 (6.7)	14,687 (5.5)	11,871 (8.4)	5048 (8.5)	
Jul, 2014~Dec, 2014	40,380 (8.7)	14,769 (5.6)	19,063 (13.5)	6548 (11.0)	
Jan, 2015~Jun, 2015	40,248 (8.6)	13,391 (5.0)	20,191 (14.3)	6666 (11.2)	
Jul, 2015~Dec, 2015	40,451 (8.7)	13,188 (5.0)	20,902 (14.8)	6361 (10.7)	
Jan, 2016~Jun, 2016	30,672 (6.6)	10,364 (3.9)	15,488 (11.0)	4820 (8.1)	
**Region** ^a,b,c^					<0.001
Southwest	79,040 (17.0)	60,507 (22.8)	11,452 (8.1)	7081 (11.9)	
Northwest	32,790 (7.0)	29,397 (11.1)	2047 (1.5)	1346 (2.3)	
South	120,146 (25.8)	80,841 (30.4)	23,049 (16.4)	16,256 (27.4)	
Northeast	13,321 (2.8)	1703 (0.6)	10,685 (7.6)	933 (1.6)	
Central	64,177 (13.8)	32,475 (12.2)	24,522 (17.4)	7180 (12.1)	
North	38,622 (8.3)	21,309 (8.0)	12,712 (9.0)	4601 (7.8)	
East	117,878 (25.3)	39,491 (14.9)	56,420 (40.0)	21,967 (37.0)	
Missing	59 (0.0)	51 (0.0)	8 (0.0)	0 (0.0)	

DOG: depressants only users. SOG: stimulants only users. DSG: both depressants and stimulants users. ^a^ refers to a difference between DOG and SOG that was significant (*p* < 0.001). ^b^ refers to a difference between DOG and DSG that was significant (*p* < 0.001). ^c^ refers to a difference between SOG and DSG that was significant (*p* < 0.001). ^d^ One-way ANOVA was used to compare the age among the three groups.

**Table 2 ijerph-17-05483-t002:** Comparison of HIV infection rates by characteristics among opioid-type only, synthetic-type only and both opioid-type and synthetic-type drug users.

Characteristics	DOG			SOG			DSG			AOR (95%CI)	AOR (95%CI)	AOR (95%CI)
	HIV	N	Rate (95%CI)	HIV	N	Rate (95%CI)	HIV	N	Rate (95%CI)	(DOG vs SOG)	(DSG vs SOG)	(DOG vs DSG)
**Total**	7906	26,5774	2.97 (2.91–3.04)	635	140,895	0.45 (0.42–0.49)	981	59,364	1.65 (1.55–1.76)	2.07 (1.88–2.28) ***	1.95 (1.72–2.22) ***	1.28 (1.19–1.37) ***
**Sex**												
Male	7078	24,0730	2.94 (2.87–3.01)	521	117,297	0.44 (0.41–0.48)	832	51,765	1.61 (1.50–1.72)	2.17 (1.95–2.40) ***	1.91 (1.67–2.20) ***	1.32 (1.22–1.42) ***
Female	828	25,044	3.31 (3.09–3.54)	114	23,598	0.48 (0.40–0.58)	149	7599	1.96 (1.67–2.29)	1.60 (1.24–2.07) ***	2.18 (1.59–2.99) ***	1.08 (0.89–1.31)
**Age**												
≤17	11	911	1.21 (0.68–2.02)	2	1564	0.13 (0.04–0.36)	0	137	0 (-)	1.79 (0.18–17.77)	-	-
18~24	405	23,946	1.69 (1.53–1.86)	137	32,951	0.42 (0.35–0.49)	39	4636	0.84 (0.62–1.13)	1.66 (1.28–2.15) ***	1.35 (0.88–2.05)	1.27 (0.90–1.81)
25~44	6406	20,1105	3.19 (3.11–3.26)	435	95,836	0.45 (0.41–0.50)	727	44,416	1.64 (1.52–1.76)	2.24 (2.00–2.51) ***	1.92 (1.66–2.23) ***	1.34 (1.24–1.46) ***
45~	1082	39,786	2.72 (2.56–2.88)	60	10,528	0.57 (0.44–0.72)	215	10,172	2.11 (1.85–2.41)	1.59 (1.17–2.15) **	2.67 (1.91–3.74) ***	1.00 (0.86–1.18)
Missing	2	26	7.69 (2.38–21.43)	1	16	6.25 (1.51–23.06)	0	3	0 (-)	1.25 (0.10–15.01) ^a^	-	-
**Ethnicity**												
Han	5040	215,647	2.34 (2.27–2.40)	467	119,358	0.39 (0.36–0.43)	761	52,483	1.45 (1.35–1.55)	1.72 (1.53–1.92) ***	1.87 (1.62–2.16) ***	1.18 (1.09–1.28) ***
Minority	2520	35,014	7.20 (6.92–7.48)	68	7485	0.91 (0.72–1.14)	158	4199	3.76 (3.22–4.37)	3.96 (3.04–5.15) ***	2.13 (1.51–3.01) ***	1.65 (1.39–1.97) ***
Missing	346	15,113	2.29 (2.06–2.54)	100	14,052	0.71 (0.59–0.86)	62	2682	2.31 (1.81–2.92)	2.23 (1.67–2.97) ^***^	2.35 (1.61–3.41) ***	1.09 (0.81–1.45)
**Education**												
High school or above	477	24,874	1.92 (1.75–2.09)	88	21,977	0.40 (0.33–0.49)	69	7151	0.96 (0.76–1.21)	1.27 (0.95–1.71)	1.51 (1.00–2.28)	1.09 (0.83–1.44)
Junior high school	3847	15,4243	2.49 (2.42–2.57)	400	94,789	0.42 (0.38–0.46)	564	38,280	1.47 (1.36–1.60)	1.83 (1.61–2.07) ***	1.98 (1.68–2.33) ***	1.18 (1.07–1.29) **
Primary school or below	3261	77,349	4.22 (4.07–4.36)	134	21,666	0.62 (0.52–0.73)	329	12,871	2.56 (2.29–2.84)	2.75 (2.28–3.32) ***	2.06 (1.62–2.63) ***	1.38 (1.22–1.55) ***
Missing	321	9308	3.45 (3.09–3.84)	13	2463	0.53 (0.31–0.85)	19	1062	1.79 (1.15–2.68)	3.06 (1.66–5.67) ***	2.70 (1.15–6.33) **	1.46 (0.88–2.44)
**Marital status**												
Married	3325	100,079	3.32 (3.21–3.44)	348	76,786	0.45 (0.41–0.50)	444	24,236	1.83 (1.67–2.01)	2.10 (1.84–2.41) ***	2.04 (1.70–2.44) ***	1.32 (1.18–1.46) ***
Unmarried	3571	135,569	2.63 (2.55–2.72)	240	52,955	0.45 (0.40–0.51)	415	28,659	1.45 (1.32–1.59)	1.96 (1.68–2.27) ***	1.76 (1.45–2.14) ***	1.28 (1.15–1.43) ***
Divorced or Widowed	735	22,748	3.23 (3.01–3.47)	40	9592	0.42 (0.31–0.56)	116	5850	1.98 (1.65–2.36)	2.11 (1.46–3.06) ***	2.40 (1.55–3.72) ***	1.05 (0.85–1.31)
Missing	275	7378	3.73 (3.31–4.18)	7	1562	0.45 (0.22–0.84)	6	619	0.97 (0.45–1.89)	3.40 (1.52–7.61) **	2.28 (0.65–8.04)	2.33 (1.02–5.34) *
**Injection drug use**												
No	759	59,374	1.28 (1.19–1.37)	233	65,330	0.36 (0.31–0.40)	228	21,833	1.04 (0.92–1.18)	1.86 (1.54–2.23) ***	2.14 (1.74–2.62) ***	0.95 (0.81–1.12)
Yes	4635	87,142	5.32 (5.17–5.47)	24	726	3.31 (2.23–4.75)	472	15,249	3.10 (2.83–3.38)	1.38 (0.90–2.09)	0.81 (0.53–1.26)	1.55 (1.40–1.72) ***
Not sure	2512	119,258	2.11 (2.03–2.19)	378	74,839	0.51 (0.46–0.56)	281	22,282	1.26 (1.12–1.41)	2.39 (2.10–2.72) ***	1.90 (1.59–2.26) ***	1.23 (1.08–1.40) **
**Methadone treatment**												
No	6257	229,374	2.73 (2.66–2.80)	635	140,895	0.45 (0.41–0.48)	759	50,169	1.51 (1.41–1.62)	2.18 (1.97–2.40) ***	1.94 (1.70–2.21) ***	1.26 (1.16–1.36) ***
Yes	1649	36,400	4.53 (4.32–4.75)	0	0	-	222	9195	2.41 (2.12–2.74)	-	-	1.33 (1.14–1.54) ***
**Date of HIV test**												
Jan, 2008~Jun, 2008	80	2078	3.85 (3.10–4.74)	0	116	0 (-)	0	72	0 (-)	-	-	-
Jul, 2008~Dec, 2008	673	21,040	3.20 (2.97–3.44)	3	2045	0.15 (0.05–0.35)	13	1669	0.78 (0.46–1.26)	9.08 (2.88–28.64) ***	5.36 (1.40–20.56) *	2.07 (1.17–3.65) *
Jan, 2009~ Jun, 2009	792	24,030	3.30 (3.07–3.53)	6	1877	0.32 (0.15–0.62)	10	1658	0.60 (0.33–1.03)	3.16 (1.37–7.30) **	1.27 (0.39–4.11)	2.32 (1.22–4.40) *
Jul, 2009~ Dec, 2009	626	24,894	2.51 (2.33–2.72)	9	4416	0.20 (0.11–0.36)	12	2533	0.47 (0.27–0.78)	2.83 (1.42–5.65) **	1.74 (0.60–5.12)	1.97 (1.10–3.54) *
Jan, 2010~ Jun, 2010	427	20936	2.04 (1.86–2.24)	13	4150	0.31 (0.18–0.51)	22	2591	0.85 (0.56–1.24)	1.47 (0.82–2.65)	1.86 (0.81–4.25)	1.14 (0.73–1.79)
Jul, 2010~ Dec, 2010	395	18,062	2.19 (1.98–2.41)	12	4176	0.29 (0.17–0.47)	18	2435	0.74 (0.47–1.12)	2.09 (1.14–3.81) *	0.71 (0.27–1.85)	1.64 (1.01–2.68) *
Jan, 2011~ Jun, 2011	364	16,181	2.25 (2.03–2.49)	13	3846	0.34 (0.20–0.55)	20	2391	0.84 (0.54–1.24)	2.97 (1.60–5.53) **	1.92 (0.81–4.55)	1.95 (1.22–3.12) **
Jul, 2011~ Dec, 2011	304	13,425	2.26 (2.02–2.53)	9	3388	0.27 (0.14–0.47)	23	2262	1.02 (0.68–1.47)	2.36 (1.16–4.79) *	3.05 (1.26–7.42) *	1.34 (0.86–2.10)
Jan, 2012~ Jun, 2012	352	14,519	2.42 (2.18–2.68)	15	5091	0.29 (0.18–0.46)	24	2802	0.86 (0.58–1.23)	2.89 (1.65–5.05) ***	1.49 (0.65–3.38)	1.89 (1.22–2.91) **
Jul, 2012~ Dec, 2012	350	13,360	2.62 (2.36–2.90)	20	5595	0.36 (0.23–0.53)	29	2943	0.99 (0.69–1.38)	1.72 (1.03–2.87) *	1.12 (0.53–2.38)	1.64 (1.01–2.43) *
Jan, 2013~ Jun, 2013	372	14,420	2.58 (2.33–2.85)	25	8207	0.30 (0.21–0.44)	45	4127	1.09 (0.82–1.43)	2.00 (1.26–3.16) **	1.87 (1.04–3.38) *	1.46 (1.06–2.02) *
Jul, 2013~ Dec, 2013	514	16,439	3.13 (2.87–3.40)	39	10,473	0.37 (0.27–0.50)	72	4438	1.62 (1.29–2.02)	2.43 (1.67–3.54) ***	2.07 (1.25–3.43) **	1.32 (1.01–1.71) *
Jan, 2014~ Jun, 2014	422	14,687	2.87 (2.61–3.15)	52	11,871	0.44 (0.33–0.56)	78	5048	1.55 (1.24–1.91)	1.57 (1.11–2.23) *	1.83 (1.19–2.79) **	1.11 (0.86–1.44)
Jul, 2014~ Dec, 2014	606	14,769	4.1 (3.79–4.44)	98	19,063	0.51 (0.42–0.62)	129	6548	1.97 (1.66–2.32)	2.32 (1.77–3.04) ***	1.73 (1.23–2.44) **	1.60 (1.30–1.95) ***
Jan, 2015~ Jun, 2015	587	13,391	4.38 (4.04–4.75)	82	20,191	0.41 (0.33–0.50)	148	6666	2.22 (1.89–2.59)	2.85 (2.13–3.81) ***	2.28 (1.61–3.22) ***	1.40 (1.15–1.70) **
Jul, 2015~ Dec, 2015	561	13188	4.25 (3.92–4.61)	155	20,902	0.74 (0.63–0.86)	207	6361	3.25 (2.84–3.71)	2.37 (1.87–3.00) ***	2.56 (1.97–3.33) ***	1.03 (0.87–1.22)
Jan, 2016~ Jun, 2016	481	10364	4.64 (4.24–5.06)	84	15,488	0.54 (0.44–0.66)	131	4820	2.72 (2.29–3.20)	2.83 (2.11–3.78) ***	1.73 (1.19–2.51) **	1.42 (1.16–1.75) **
**Region**												
Southwest	3589	60,507	5.93 (5.74–6.13)	114	11,452	1.00 (0.83–1.19)	300	7081	4.24 (3.78–4.73)	2.47 (2.03–3.01) ***	2.52 (1.96–3.25) ***	1.36 (1.20–1.54) ***
Northwest	812	29,397	2.76 (2.58–2.96)	25	2047	1.22 (0.83–1.74)	29	1346	2.15 (1.50–3.01)	1.15 (0.75–1.77)	1.00 (0.51–1.93)	0.94 (0.63–1.40)
South	2142	80,841	2.65 (2.54–2.76)	147	23,049	0.64 (0.54–0.74)	380	16256	2.34 (2.11–2.58)	2.12 (1.75–2.55) ***	1.99 (1.58–2.50) ***	1.23 (1.09–1.39) **
Northeast	102	1703	5.99 (4.94–7.21)	21	10,685	0.20 (0.13–0.29)	39	933	4.18 (3.06–5.59)	7.69 (3.21–18.38) ***	4.32 (1.58–11.86) **	1.33 (0.88–2.00)
Central	464	32,475	1.43 (1.3–1.56)	155	24,522	0.63 (0.54–0.74)	122	7180	1.70 (1.42–2.01)	3.40 (2.66–4.34) ***	2.60 (1.95–3.48) ***	1.29 (1.04–1.61) *
North	281	21,309	1.32 (1.17–1.48)	54	12,712	0.42 (0.33–0.55)	36	4601	0.78 (0.57–1.06)	2.10 (1.49–2.95) ***	1.00 (0.57–1.75)	1.59 (1.10–2.30) *
East	516	39,491	1.31 (1.2–1.42)	119	56,420	0.21 (0.18–0.25)	75	21,967	0.34 (0.27–0.42)	3.04 (2.33–3.96) ***	1.10 (0.73–1.64)	2.46 (1.90–3.19) ***
Missing	0	51	0(-)	0	8	0(-)	0	0	-	-	-	-

DOG: depressants only users. SOG: stimulants only users. DSG: both depressants and stimulants users. AOR adjusted for sex, age, ethnicity, education, marital status, injection drug use, methadone treatment, date of HIV test and region. * suggested *p* < 0.05, ** suggested *p* < 0.01, *** suggested *p* < 0.001.

**Table 3 ijerph-17-05483-t003:** Associated characteristics of HIV infection among depressants only, stimulants only and both depressants and stimulants users.

Characteristics	DOG		SOG		DSG	
	OR (95% CI)	AOR (95% CI)	OR (95% CI)	AOR (95% CI)	OR (95% CI)	AOR (95% CI)
**Sex**						
Male	1	1	1	1	1	1
Female	1.13 (1.05–1.21) **	1.17 (1.08–1.26) ***	1.09 (0.89–1.33)	1.34 (1.09–1.66) **	1.22 (1.03–1.46) *	1.55 (1.29–1.86) ***
**Age when first HIV test**						
≤17	1	1	1	1	-	-
18~24	1.41 (0.77–2.57)	1.66 (0.90–3.04)	3.26 (0.81–13.18)	3.82 (0.94–15.47)	1	1
25~44	2.69 (1.48–4.88) **	3.78 (2.07–6.89) ***	3.56 (0.89–14.30)	4.90 (1.21–19.80) *	1.96 (1.42–2.71) ***	2.18 (1.56–3.04) ***
45~	2.29 (1.26–4.16) **	3.69 (2.01–6.76) ***	4.48 (1.09–18.33) *	7.20 (1.73–29.89) **	2.55 (1.81–3.59) ***	3.04 (2.10–4.39) ***
Missing	6.82 (1.43–32.45) *	4.84 (0.99–23.58)	52.07 (4.48–605.62) **	30.63 (2.53–370.29) **	-	-
**Ethnicity**						
Han	1	1	1	1	1	1
Minority	3.24 (3.09–3.40) ***	2.87 (2.72–3.02) ***	2.33 (1.81–3.01) ***	1.68 (1.29–2.20) ***	2.66 (2.23–3.16) ***	1.87 (1.56–2.25) ***
Missing	0.98 (0.88–1.09)	1.11 (0.99–1.24)	1.83 (1.47–2.27) ***	1.32 (1.05–1.65) *	1.61 (1.24–2.09) ***	1.42 (1.08–1.86) *
**Education**						
High school or above	1	1	1	1	1	1
Junior high school	1.31 (1.19–1.44) ***	1.34 (1.22–1.48) ***	1.05 (0.84–1.33)	1.05 (0.83–1.33)	1.54 (1.19–1.97) **	1.41 (1.09–1.82) **
Primary school or below	2.25 (2.04–2.48) ***	1.89 (1.70–2.09) ***	1.55 (1.18–2.03) **	1.35 (1.03–1.78) *	2.69 (2.07–3.50) ***	1.97 (1.50–2.58) ***
Missing	1.83 (1.58–2.11) ***	1.75 (1.28–2.39) ***	1.32 (0.74–2.37)	1.07 (0.46–2.49)	1.87 (1.12–3.12) *	2.01 (1.08–3.73) *
**Marital status**						
Married	1	1	1	1	1	1
Unmarried	0.79 (0.75–0.83) ***	0.83 (0.79–0.87) ***	1.00 (0.85–1.18)	0.98 (0.81–1.17)	0.79 (0.68–0.90) **	0.82 (0.72–0.95) **
Divorced or Widowed	0.97 (0.90–1.05)	0.92 (0.84–1.00)	0.92 (0.66–1.28)	0.94 (0.67–1.33)	1.08 (0.88–1.33)	1.06 (0.85–1.32)
Missing	1.13 (0.99–1.28)	1.22 (0.88–1.68)	0.99 (0.47–2.09)	1.94 (0.61–6.14)	0.52 (0.23–1.18)	0.82 (0.29–2.27)
**Injection drug use**						
No	1	1	1	1	1	1
Yes	4.34 (4.02–4.69) ***	4.96 (4.58–5.37) ***	9.55 (6.23–14.64) ***	7.42 (4.79–11.48) ***	3.03 (2.58–3.55) ***	2.68 (2.28–3.17) ***
Not sure	1.66 (1.53–1.80) ***	1.53 (1.40–1.66) ***	1.42 (1.20–1.67) ***	1.16 (0.97–1.38)	1.21 (1.02–1.44) *	0.96 (0.80–1.15)
**Methadone treatment**						
No	1	1	1	1	1	1
Yes	1.69 (1.60–1.79) ***	1.80 (1.69–1.91) ***	-	-	1.61 (1.39–1.87) ***	1.42 (1.21–1.67) ***
**Date of HIV test**						
Jan, 2008~Jun, 2008	1	1	-	-	-	-
Jul, 2008~Dec, 2008	0.83 (0.65–1.05)	0.79 (0.62–1.01)	1	1	1	1
Jan, 2009~Jun, 2009	0.85 (0.67–1.08)	0.80 (0.63–1.02)	2.18 (0.55–8.74)	2.32 (0.58–9.30)	0.77 (0.34–1.77)	0.83 (0.36–1.91)
Jul, 2009~Dec, 2009	0.64 (0.51–0.82) ***	0.68 (0.53–0.87) **	1.39 (0.38–5.14)	1.50 (0.40–5.55)	0.61 (0.28–1.33)	0.65 (0.29–1.45)
Jan, 2010~Jun, 2010	0.52 (0.41–0.66) ***	0.51 (0.40–0.66) ***	2.14 (0.61–7.51)	2.39 (0.68–8.45)	1.09 (0.55–2.17)	1.12 (0.55–2.25)
Jul, 2010~Dec, 2010	0.56 (0.44–0.71) ***	0.57 (0.44–0.73) ***	1.96 (0.55–6.96)	2.08 (0.58–7.45)	0.95 (0.46–1.94)	0.89 (0.43–1.85)
Jan, 2011~Jun, 2011	0.58 (0.45–0.74) ***	0.55 (0.43–0.71) ***	2.31 (0.66–8.11)	2.53 (0.71–9.04)	1.08 (0.53–2.17)	0.93 (0.46–1.90)
Jul, 2011~Dec, 2011	0.58 (0.45–0.74) ***	0.50 (0.39–0.65) ***	1.81 (0.49–6.70)	2.08 (0.55–7.80)	1.31 (0.66–2.59)	1.09 (0.54–2.19)
Jan, 2012~Jun, 2012	0.62 (0.49–0.80) ***	0.55 (0.43–0.71) ***	2.01 (0.58–6.96)	2.18 (0.62–7.66)	1.10 (0.56–2.17)	0.87 (0.43–1.74)
Jul, 2012~Dec, 2012	0.67 (0.53–0.86) **	0.57 (0.44–0.74) ***	2.44 (0.73–8.23)	2.77 (0.81–9.50)	1.27 (0.66–2.45)	0.93 (0.47–1.82)
Jan, 2013~Jun, 2013	0.66 (0.52–0.85) **	0.53 (0.41–0.68) ***	2.08 (0.63–6.90)	2.33 (0.69–7.86)	1.40 (0.76–2.61)	0.96 (0.51–1.82)
Jul, 2013~Dec, 2013	0.81 (0.63–1.03)	0.62 (0.48–0.79) ***	2.54 (0.79–8.24)	2.76 (0.84–9.08)	2.10 (1.16–3.80) *	1.28 (0.70–2.37)
Jan, 2014~Jun, 2014	0.74 (0.58–0.94) *	0.54 (0.42–0.69) ***	3.00 (0.93–9.60)	3.22 (0.99–10.51)	2.00 (1.11–3.61) *	1.24 (0.68–2.28)
Jul, 2014~Dec, 2014	1.07 (0.84–1.36)	0.82 (0.64–1.05)	3.52 (1.11–11.11) *	3.80 (1.18–12.21) *	2.56 (1.44–4.54) **	1.49 (0.82–2.69)
Jan, 2015~Jun, 2015	1.15 (0.90–1.45)	0.81 (0.63–1.04)	2.78 (0.88–8.79)	3.07 (0.95–9.89)	2.89 (1.64–5.11) ***	1.55 (0.86–2.80)
Jul, 2015~Dec, 2015	1.11 (0.87–1.41)	0.86 (0.67–0.11)	5.09 (1.62–15.95) **	4.75 (1.49–15.20) **	4.29 (2.44–7.52) ***	2.21 (1.24–3.97) **
Jan, 2016~Jun, 2016	1.22 (0.96–1.55)	0.96 (0.75–1.24)	3.71 (1.17–11.75) *	3.90 (1.21–12.58) *	3.56 (2.01–6.31) ***	1.76 (0.97–3.17)
**Region**						
Southwest	1	1	1	1	1	1
Northwest	0.45 (0.42–0.49) ***	0.50 (0.46–0.54) ***	1.23 (0.80–1.90)	1.21 (0.78–1.88)	0.50 (0.34–0.73) ***	0.60 (0.40–0.88) *
South	0.43 (0.41–0.46) ***	0.43 (0.40–0.45) ***	0.64 (0.50–0.82) ***	0.70 (0.55–0.91) **	0.54 (0.46–0.63) ***	0.64 (0.54–0.75) ***
Northeast	1.01 (0.83–1.24)	0.83 (0.67–1.02)	0.20 (0.12–0.31) ***	0.19 (0.12–0.31) ***	0.99 (0.70–1.39)	0.90 (0.63–1.28)
Central	0.23 (0.21–0.25) ***	0.20 (0.18–0.22) ***	0.63 (0.50–0.81) ***	0.63 (0.49–0.82) ***	0.39 (0.32–0.48) ***	0.44 (0.35–0.55) ***
North	0.21 (0.19–0.24) ***	0.29 (0.25–0.32) ***	0.42 (0.31–0.59) ***	0.45 (0.32–0.62) ***	0.18 (0.13–0.25) ***	0.22 (0.15–0.31) ***
East	0.21 (0.19–0.23) ***	0.20 (0.18–0.22) ***	0.21 (0.16–0.27) ***	0.25 (0.19–0.33) ***	0.08 (0.06–0.10) ***	0.10 (0.08–0.13) ***
Missing	-	-	-	-	-	-

DOG: depressants only users. SOG: stimulants only users. DSG: both depressants and stimulants users. AOR adjusted for sex, age, ethnicity, education, marital status, injection drug use, methadone treatment, date of HIV test and region. * suggested *p* < 0.05, ** suggested *p* < 0.01, *** suggested *p* < 0.001.

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
