# Peer review of "Epidemics of HIV Infection among Heavy Drug Users of Depressants Only, Stimulants Only, and Both Depressants and Stimulants in Mainland China: A Series, Cross-Sectional Studies"

_ijerph, 2020, doi:10.3390/ijerph17155483_

Round 1
Reviewer 1 Report
Comments to 857025
This manuscript focuses on examining differences of socio-demographic and geographic characteristics across three subgroups (depressants only, stimulants only, and both of depressants/ stimulants) among individuals who use drugs in China using bivariate and multivariate analyses. Findings suggest that trends of HIV infection among individuals who use drugs in China have been increased since 2008. Several socio-demographic and geographic characteristics were statistically associated with HIV infection among individuals who use drugs, which may provide your readers with a more meaningful understanding of your target population in China. I see how the research topic can be valuable, adding to public health research in China. However, the content and logic related to introduction, materials, and discussion may need to be reorganized for clarity.
Specific comments are as follows:
Introduction
- In general, the writing logic throughout this manuscript should be reorganized for clarity. For example, the authors seldom used conjunction words to connect two sentences. I also found that several paragraphs are repetitive. As such, they could make your readers have difficulty in reading different levels of information.
- Moreover, your literature review should focus more on those associations between HIV infection and socio-demographic and geographic characteristics among heavy drug users in Mainland China.
Page 2, Line 54,
I understand the reason why the authors wanted to mention “group sexual intercourses” in this sentence. However, if all the individuals who are in a "group sexual intercourses" all practice safer sex, such as using PrEP and condoms, the risk of HIV transmission is still extremely low. I would suggest removing this term from this sentence.
Page 2, Line 53 & Line 65
These two lines both discuss the association between drug use and HIV infection. It would be clear if you could combine them into one paragraph.
Page 2, Line 74-82
Can you please provide your rationale for discussing respondent-driven sampling here if you did not use and even mention this sampling in the rest of your paper?
Page 2, Line 83-88
This sentence discusses the limitations of using surveillance data. It would be clear if you could combine them with research limitations on page 17.
Page 2, Line 88-90
This sentence discusses your dataset. It would be helpful if you could move it to the materials and method section.
Page 2, Line 92-93
This sentence defines heavy drug users. However, the authors mentioned the definition of heavy drug users again on page 3 (2.1 definitions). They are repetitive.
Page 3. Line 100-101
Your research purpose should be consistent with your research finding. To me, the authors not only want to know about the HIV infection trends but also examine socio-demographic and geographic characteristics of HIV infection among heavy drug users in Mainland China. It would be helpful for your readers to understand the aims of your study if you could clarify your research purpose.
Materials and Methods
- Please reorganize Materials and Methods. Some parts of this section are fine, however, it would be helpful for your readers to understand the background of the dataset (NDMCDDU) and measurement of variables in your study if you could briefly describe them in your paper under this section. The authors did selectively mention variable measurement and definition under the statistical analysis section. Yet, you may want to structure a section for defining your variables before the statistical analysis section. For example, how did the dataset measure methadone maintenance treatment (MMT)? It was simply a yes/no answer but your readers only can get this information by reading your tables.
- This way, the authors would leave the statistical analysis section solely addressing data analysis.
Results
It is hard to read your Table 2. The authors may want to edit Table 2 to clearly illustrate your information. It is fine to use two tables to present information on the current Table 2.
Page 4, Line 181, 3.1 trends and risk factors of HIV infection.
- All your independent variables are socio-demographic and geographic characteristics. It would be inappropriate if you labelled those characteristics as risk factors. For example, when you found that females and people with unmarried status were associated with higher odds of HIV infection, it does not mean that females or unmarried individuals are at a higher risk of HIV infection. Your results just showed that individuals who use drugs and live with HIV are mostly from vulnerable populations, such as female, unmarried, minority, and less educated.
- Your research findings might be scaled up if you could address those significant characteristics with proper labelling and interpretations.
Page 14, Figure 3
Taiwan should not be included in the map of Mainland China due to these two countries utilizing different systems regarding public health and social welfare services to their citizens.
Discussion
Page 16, Line 272-273,
This paragraph is not even fairly describing your findings. A better statement may be “Those people who are living with HIV and are heavily using drugs were associated with being woman, non-married, older, undereducated, and racial minorities.” A meaningful discussion would be required to show your interesting findings here.
Page 16, Line 300-315
This section talks about MSM and HIV, however, sexual orientation was not included in your dataset and data analysis. It could likely make your readers feel confused about this discussion. You may just want to focus discussion section around your research findings which is the vulnerability of people living with HIV who heavily use drugs in China. This is your main selling point. Your discussion may be much clear if you could centre your discussion more on gender and social status inequality among your target population, instead of spending a paragraph discussing MSM and HIV.
Page 17, Line 320-321 This statement is redundant.
Page 17, Line 324-325
This manuscript would be strengthened if you could provide some examples of implications for public health practices based on findings of your research if you would like your readers to have some lessons learned after reading your paper.
Conclusion
The content of the current conclusion section is not even relevant and consistent with your research findings. Your conclusion would be much clear and coherent to your research findings if you could briefly mention gender, social status, and maybe geographic inequality among your target population. And this could make your research more meaningful. I understand the association between drug use and sexual orientation is considered commonly accepted, however, your research findings did not support this association.
Reviewer 2 Report
This article by Bo Zhang et al. describe the dynamics of the HIV infections in mainland China from 2008 to 2016. This article is well written but deserve revision to be acceptable for publication.
Major comments :
The choix for the interval of analysis (2008-2016) has to be fully explained. Has the number of patient to include determined ? if yes, give values, if not explain the reason.
Ethical issues : even if the study is a population-level study, ethical approval are needed to be included. Are the patients informed of their inclusion in the database? are they informed of the study?
As data are available, description of data from drug used (other than depressants and stimulants) have to be added and discussed (in supplementary maybe?).
Table 1 : could the author justify the temporal stratification by semester.
As this article could be compared to other countries that describe quasi-similar data. Please complete the discussion
Minor comments :
Table 1 : Age +/-IQR could be more interesting than +/-s.
Table 1 : Indicate p-values and not X² value
Part of table 2 (on page 7 and 8 are illisible or missing). please modify
Have the authors obtained data on the respective profession of socio-professional category of the IVD users? If yes, some information could be of interest.
Could the authors give information about the economic situation of all Chinese region, for non-chinese readers? This data has to be discussed regarding to the evolution of epidemics.
Reviewer 3 Report
In this manuscript, authors examined correlation between drug users which were grouped into DOG, SOG, and DSG, and HIV infection in China.
Using epidemic analysis, they reported that HIV infection rate was higher in users in DOG than those in SOG and DSG. Moreover, the HIV infection rate of DOG users showed a U-shaped trend, while that of SOG and DSG users revealed consistent increases. From these findings, they warn to be necessary of reassessment about the effectiveness of current strategies on HIV prevention and control among drug users. Although authors provide very important message on HIV prevention among drug users in China, this manuscript seems to lack a direct discussion why HIV infection rate is high in DOG and SOG users.
1) Authors must discuss in detail why DOG users was higher in HIV infection rate and inflectional rate of DOG users showed the U-shaped trend.
2) Most of readers would like to know why and how DOG and SOD users are related to the high risk of HIV infection. Authors must discuss in detail these in the Discussion section.
Round 2
Reviewer 1 Report
The authors have addressed most of my comments in details. I am very happy to review the current version of your manuscript. You have done a wonderful job.
Reviewer 2 Report
Thanks for the complete response you send regarding to my previous comments.
After reading them, and the correction you did on the manuscript. I think that the manuscript is suitable for publication.
Reviewer 3 Report
The authors performed all required changes in this submission.